# Predictive Factors for Difficult Laparoscopic Cholecystectomies in Acute Cholecystitis

**DOI:** 10.3390/diagnostics14030346

**Published:** 2024-02-05

**Authors:** Paul Lorin Stoica, Dragos Serban, Dan Georgian Bratu, Crenguta Sorina Serboiu, Daniel Ovidiu Costea, Laura Carina Tribus, Catalin Alius, Dan Dumitrescu, Ana Maria Dascalu, Corneliu Tudor, Laurentiu Simion, Mihail Silviu Tudosie, Meda Comandasu, Alexandru Cosmin Popa, Bogdan Mihai Cristea

**Affiliations:** 1Doctoral School, Carol Davila University of Medicine and Pharmacy Bucharest, 020021 Bucharest, Romania; paul.stoica@drd.umfcd.ro; 2Faculty of Medicine, Carol Davila University of Medicine and Pharmacy Bucharest, 020021 Bucharest, Romania; sorina.serboiu@umfcd.ro (C.S.S.); catalin.alius@umfcd.ro (C.A.); dan.dumitrescu@umfcd.ro (D.D.); ana.dascalu@umfcd.ro (A.M.D.); corneliu.tudor@umfcd.ro (C.T.); laurentiu.simion@umfcd.ro (L.S.); mihail.tudosie@umfcd.ro (M.S.T.); bogdan.cristea@umfcd.ro (B.M.C.); 3Fourth General Surgery Department, Emergency University Hospital Bucharest, 050098 Bucharest, Romania; meda.comandasu@gmail.com; 4Faculty of Medicine, University Lucian Blaga Sibiu, 550169 Sibiu, Romania; 5Department of Surgery, Emergency County Hospital Sibiu, 550245 Sibiu, Romania; 6Faculty of Medicine, Ovidius University Constanta, 900470 Constanta, Romania; 7General Surgery Department, Emergency County Hospital Constanta, 900591 Constanta, Romania; 8Faculty of Dental Medicine, Carol Davila University of Medicine and Pharmacy Bucharest, 020021 Bucharest, Romania; laura.tribus@umfcd.ro; 9Department of Internal Medicine, Ilfov Emergency Clinic Hospital Bucharest, 022104 Bucharest, Romania; 10Department of Surgical Oncology, Institute of Oncology “Prof. Dr. Al. Trestioreanu”, 022328 Bucharest, Romania; 11Department of General Surgery, Colentina Clinic Hospital, 020125 Bucharest, Romania

**Keywords:** acute cholecystitis, difficult laparoscopic cholecystectomies, prediction, biomarkers, outcomes

## Abstract

Laparoscopic cholecystectomy (LC) is the gold standard treatment in acute cholecystitis. However, one in six cases is expected to be difficult due to intense inflammation and suspected adherence to and involvement of adjacent important structures, which may predispose patients to higher risk of vascular and biliary injuries. In this study, we aimed to identify the preoperative parameters with predictive value for surgical difficulties. A retrospective study of 255 patients with acute cholecystitis admitted in emergency was performed between 2019 and 2023. Patients in the difficult laparoscopic cholecystectomy (DLC) group experienced more complications compared to the normal LC group (33.3% vs. 15.3%, *p* < 0.001). Age (*p* = 0.009), male sex (*p* = 0.03), diabetes (*p* = 0.02), delayed presentation (*p* = 0.03), fever (*p* = 0.004), and a positive Murphy sign (*p* = 0.007) were more frequently encountered in the DLC group. Total leukocytes, neutrophils, and the neutrophil-to-lymphocyte ratio (NLR) were significantly higher in the DLC group (*p* < 0.001, *p* = 0.001, *p* = 0.001 respectively). The Tongyoo score (AUC ROC of 0.856) and a multivariate model based on serum fibrinogen, thickness of the gallbladder wall, and transverse diameter of the gallbladder (AUC ROC of 0.802) showed a superior predictive power when compared to independent parameters. The predictive factors for DLC should be assessed preoperatively to optimize the therapeutic decision.

## 1. Introduction

Laparoscopic cholecystectomy (LC) is one of the most frequently performed procedures in general surgery [1,2]. It has become the gold standard of treatment in acute and chronic cholecystitis, aiming to ensure disability-free survival with relief of symptoms [3].

While most LCs are performed safely on an elective basis, even in day care centers [4], performing surgery in acute cholecystitis may be challenging due to inadequate visualization of anatomical landmarks and the impossibility of achieving a critical view of safety (CVS). Other patient-related factors, such as previous upper abdominal surgery or obesity, may be associated with surgical challenges due to limited operative exposure [5,6].

However, the definition of a difficult laparoscopic cholecystectomy (DLC) varies widely according to the experience of the surgical team. Previous studies defined DLC as associated with prolonged operation time (>180 min), increased bleeding (>300 mL), urgent need for involvement of a more experienced surgeon, the use of “bailout” procedures, or conversion to open surgery [5,6,7,8]. In a study by Anees et al. [9], DLC was defined as LC exceeding 60 min or damage of the cystic artery before ligation or clipping. In the search for a consensus involving the definition of “difficult” LC, Alba Manuel-Vázquez et al. performed a Delphi study of Spanish surgeons with more than 10 years of experience in LC [10]. They concluded, based on a majority consensus of greater than or equal to 80%, that DLC should be considered in the presence of any one of the following intraoperative conditions: non-evident anatomical visualization, severe inflammation of the Calot triangle, bile duct injury, conversion to laparotomy, Mirizzi syndrome, scleroatrophic gallbladder, or pericholecystic abscess [10]. 

Several score systems that combined clinical, biological, and imaging parameters were designed to increase the accuracy of preoperative identification of difficult cholecystectomies, including both elective and emergency operated cases [11,12,13,14,15,16,17], combining personal history-related factors with clinical and imaging parameters. Ramirez-Giraldo et al. [12] comparatively evaluated the accuracy of seven scoring systems in predicting DLC in the same group of patients that underwent LC and found the best predictive value for the Tongyoo predictive model. However, these models were designed based on study groups including both elective and emergency LC. Preoperative prediction of DLC is particularly important for therapeutic management and decreases the incidence of intraoperative and postoperative complications, especially biliary and vascular lesions. The 2020 World Society of Emergency Surgery (WSES) guidelines for the detection and management of bile duct injury during cholecystectomy [18] recommended a comprehensive preoperative evaluation to detect at-risk conditions, choose the best surgical approach, and provide appropriate patient disclosure of the possible risk for complications or conversion to open surgery [19,20].

The present study aims to investigate the biological and imaging preoperative factors with predictive value for DLC in patients specifically with acute cholecystitis and to compare their predictive value to the Tongyoo scoring system [11].

## 2. Materials and Methods

We conducted a retrospective study over 4 years, June 2019–June 2023, on patients with acute cholecystitis admitted in emergency in the 4th Department of Surgery, Emergency University Hospital Bucharest. The diagnosis and severity grading of acute cholecystitis were performed according to Tokyo Guidelines TG13/18, consisting of local signs of inflammation in the right upper quadrant (RUQ); systemic signs of inflammation, including fever and elevated white blood cells (WBC) and/or C-reactive protein (CRP); along with ultrasound findings suggestive of acute calculous cholecystitis [16]. All patients underwent surgery by the same operating team. The standard protocol included initiation of broad spectrum intravenous antibiotic therapy immediately after admission. In mild cases, intravenous ceftriaxone (1 g/12 h) was used, while in medium and severe cases, a combination of ceftriaxone or piperacillin/tazobactam (4 g + 0.5 g/8 h) and metronidazole (1 g/12 h) was administered for a minimum 24–48 h after surgery, according to clinical outcome. In cases with empiema of the gallbladder, parietal micro-abscesses, or pericholecystic abscess, antibiotic therapy was adjusted if necessary, according to the results of bile microbiological exam and antibiogram. Thrombosis prophylaxis was achieved by administering low-molecular-weight heparin in the perioperative period, adjusting the dose according to m\body mass and comorbidities. Patients with ASA of 3 or more underwent general supportive therapy and correction of abnormalities in electrolyte concentration rebalance.

Emergency laparoscopic cholecystectomy was performed as soon as possible to be performed safely, within a time frame of 96 h after admission. As standard protocol, we used the conventional four-trocar operative technique, with the patient placed in a supine reverse Trendelenburg position. After initial pneumoperitoneum insufflation, the surgery was performed using a 10 mm optical trocar in the umbilical region, a 10 mm operating trocar in the left subcostal region, a 5 mm operating trocar in the right lower quadrant, and a 5 mm retractor trocar in the epigastric region. Postoperative drainage of the gallbladder fossa was used in all of these patients.

The inclusion criteria were: patients aged 18 years old or older, with acute cholecystitis who underwent laparoscopic cholecystectomy (LC) in the same admission, including cases that required conversion to open surgery.

The exclusion criteria were: acalculous acute cholecystitis, patients that underwent open surgical procedures from the beginning, and the co-existence of associated pancreatitis, cholangitis, systemic inflammatory diseases, and malignancies.

Based on the specific intraoperative findings, the patients included in the study were divided into two groups: the normal LC group and the DLC group. The criteria for DLC were those established by the Spanish expert’s consensus [12]. The two study groups were compared in terms of preoperative clinical, biological, and imaging findings. Data were collected from medical sheets, operatory protocols, and electronic records of the patients. A predictive score for DLC based on the study by Tongyoo and Randhawa [11] was calculated for each patient, and the predictive value was comparatively analyzed For each patient, a combination of general, clinical and ultrasound parameters were summed. As described by Tongyoo [11], 1 point was added for age over 50 and male sex, while personal history of biliary colic and inflammation added 4 more points. In case of obese or overweight patients, a BMI of 25–27.5 was noted with 1 point, while for more than 27.5, two points were added to individual score. In clinical evaluation, a palpable or retracted gallbladder was noted with 1 point, while evidences of previous abdominal surgery was evaluated with 1 if infraumbilical, or 2 points if the supraumbilical area was involved. The ultrasound parameters included in the score were: gallbladder wall thickness equal or more than 4 mm (2 points), pericholecystic collection (1 point), and impacted stone (1 point). As designed by Tongyoo and Randhawa, a value of 6 or more should be a indicator of intraoperative difficulties, while if more than 11, a very difficult case should be expected.

Statistical analysis was performed with EasyMedStat (version 3.30.2; www.easymedstat.com) and MedCalc (version 22.016; https://www.medcalc.org/) software. Numeric variables were expressed as the mean (±SD) and discrete outcomes as absolute and relative (%) frequencies. Group comparability was assessed by comparing baseline demographic data and follow-up duration between groups. Normality and heteroskedasticity of continuous data were assessed with Shapiro–Wilk and Levene’s tests, respectively. Continuous outcomes were compared with unpaired Student *t*-test or Mann–Whitney U test according to data distribution. Discrete outcomes were compared with Chi-squared or Fisher’s exact tests. The alpha risk was set to 5% and two-tailed tests were used. Logistic regression was used to investigate the correlation between relevant biological and imaging parameters with difficult/very difficult cholecystectomies. ROC curves were employed to characterize the predictive value of the two scoring systems, as well as the most relevant parameters based on the statistical analysis, including specificity, sensitivity, positive predictive value, and negative predictive value. A regression model based on the most relevant parameters was comparatively analyzed with other two scoring systems [11,16].

## 3. Results

A total of 255 patients, aged between 19 and 90 years, were included in the study. The mean age was 57.1 ± 14.2 years in patients with DLC, which was significantly higher compared to the control group (52.4 ± 14.2 years, *p* = 0.009). The gender distribution revealed a higher male proportion in the difficult LC group (*p* = 0.032). While we found no differences in terms of obesity, previous upper abdominal surgery, and ASA risk grading between the two study groups, the rate of diabetes mellitus was higher in the difficult LC group (*p* = 0.022) (Table 1 and Appendix A).

Delayed presentation, fever, and a positive Murphy sign were more frequently encountered in the DLC group (*p* = 0.03, *p* = 0.004, and *p* = 0.007, respectively). Consistent with the clinical findings, leukocytes, neutrophils, and the neutrophil-to-lymphocyte ratio (NLR) were significantly associated with the DLC group (*p* < 0.001, *p* = 0.001, and *p* = 0.001, respectively).

An abdominal ultrasound exam was performed in all cases to confirm the existence of gallbladder lithiasis and to assess the severity of local inflammatory changes. We encountered significantly increased thickness of the gallbladder wall in the DLC group, with a mean of 6.0 (± 2.8) vs. a median value of 4.2 (± 1.5) mm in the normal LC group (*p* < 0.001). We found also significantly higher values in terms of transverse diameter and length of the gallbladder (*p* < 0.001) (Table 2 and Appendix A).

Regarding the descriptive imaging findings, the double contour of the gallbladder wall, positive ultrasound Murphy sign, as well as the presence of the pericholecystic fluid were encountered more frequently in the DLC group (*p* < 0.001, *p* = 0.002, and *p* < 0.001, respectively).

Conversion to open cholecystectomy was performed in 24 cases (22.8%) in the DLC group (Table 3).

The reasons for conversion were: biliary fistula (2 cases, 1.9%), biliary peritonitis (2 cases, 1.9%), Mirizzi syndrome (1 case, 0.9%), pericholecystic abscess (3 cases, 2.86%), gangrenous cholecystitis (7 cases, 6.6%), severe inflammation and inability to safely dissect important structures (bile duct, cystic duct, hepatic arteries, or portal venous structures) from surrounding tissues (9 cases, 8.5%).

Both mean total hospital stay (+1.7 days) and mean postoperative hospital stay (+1.5 days) were significantly higher in the DLC group (*p* < 0.001, Table 3). However, the differences between the two study groups had limited clinical significance.

Patients in the DLC group experienced more complications compared to the normal LC group (33.33% vs. 15.33%, *p* < 0.001). However, when analyzed by type, only surgical site infections appeared to be significantly higher in patients with difficult cholecystectomies. One explanation might be the number of conversions to open surgery, which favors this type of complication, compared to laparoscopic small incision.

The postoperative complications encountered in the two study groups were analyzed according to the Clavien–Dindo classification (Table 4).

While no significant differences were encountered for complications graded 4 and 5, we found a higher incidence of surgical site infections (grade 1), and surgery-related complications with either conservatory (grade 2), or interventional (grade 3) treatment in the DLC group compared to the normal LC group (*p* = 0.006, *p* = 0.03, and *p* = 0.01, respectively).

### Preoperative Prediction of Difficult LC

A logistic regression analysis was performed to assess the relationship between difficult LC and preoperative parameters.

The best predictive value was observed for a multivariate model combining serum fibrinogen (mg/dL), thickness of the gallbladder wall(mm), and transverse diameter of the gallbladder(mm). The odds ratio, upper and lower confidence intervals, and *p*-value for each variable included in the statistical model are described in Table 5.

The predictive value of different parameters was analyzed with ROC curves. Area under curve (AUC), sensitivity, specificity, positive predictive value, and negative predictive value were comparatively analyzed (Table 6).

The Tongyoo score proved to have the best predictive value among the tested variables and could be a useful tool in clinical evaluation. The multivariate model based on three variables, serum fibrinogen, gallbladder wall thickness, and transverse diameter of the gallbladder, had a lower but not statistically significant statistical value (AUC ROC of 0.802 vs. 0.857, *p* = 0.06). Both prediction models presented good specificity, according to Mandrekar’s classification of clinical tests based on AUC ROC [21] but fair sensitivity at the cut-off value. Independent parameters had significantly lower (*p* < 0.01) predictive value, and their individual importance as a predictive value was far too low for clinical practice.

## 4. Discussion

As surgical instruments and skill improved over time, the laparoscopic approach became preferred even for patients of advanced age. The current contraindications for LC are limited and may be related to general conditions (associated pathologies with high anesthetic and surgical risk) or particular local findings, such as suspicion of gallbladder carcinoma, calcified gallbladder, cholecystoenteric fistula, Mirizzi syndrome, or extensive previous surgery in the upper abdominal region [22,23].

However, challenging intraoperative situations are encountered, especially in patients undergoing LC for acute cholecystitis. Fugger et al. found that currently, 1 in 6 cholecystectomies is difficult [24]. Preoperative assessment of potential intraoperative difficulties is important to ensure an adequate therapeutic approach [25,26]. When the critical view of safety cannot be achieved, the surgical team should be prepared to use a bailout procedure or conversion to open surgery. Recently, the intraoperative use of near-infrared indocyanine green fluorescent cholangiography (NIRFC) has provided a valuable tool to shorten the operation time and reduce blood loss and the incidence of biliary injuries [27,28].

Previous studies showed that certain risk factors may predict intraoperative difficulties, such as total leukocytosis [25,29], fibrinogen [25,30], NLR [31], and C-reactive protein [18,32]. In a recent study by Anees et al., preoperative C-reactive protein (CRP) with values > 11 mg/dL was associated with the highest odds of presenting DLC [18]. Due to the lack of availability of serum CRP in an emergency, we could not investigate the predictive value of this biomarker in the present study.

In the present study, among 255 patients with acute cholecystitis who underwent emergency surgery, we found that several clinical, biological, and imaging parameters correlated well with DLC. Consistent with the findings of Di Buono et al. [25], we also found a good correlation between total leukocytes, fibrinogen, and DLC. A higher NLR value reflected the intensity of local inflammation in these cases, with an increase of proinflammatory cytokines. Other previous studies found positive correlations between NLR and complicated acute cholecystitis [33,34] and difficult LC [31] at cut-off values of 3, 4.18, and 5.65, respectively. The differences encountered may be explained by the different criteria used to define severe inflammation and difficult LC.

However, in the present study, a multivariate regression model based on serum fibrinogen, thickness of the gallbladder wall, and transverse diameter of the gallbladder showed the best value for predicting difficult cases. Interestingly, fibrinogen value better characterized the severity of local inflammation and adhesions leading to difficult dissection than total leukocytes, neutrophils, or NLR value. Fibrinogen is an acute-phase reactant that plays a key role in the acute-phase response caused by tissue injury [35]. Acute inflammation imbalances fluid–coagulant system activity to a prothrombotic state [36]. Fibrinogen and fibrin play multiple roles in the defense mechanism. Several studies showed the roles of fibrinogen in leukocyte adhesion at the endothelium and migration, although the exact mechanisms are still a subject of controversy. Moreover, fibrin deposits at the inflammation site activate leukocytes via the beta2-integrin family, increasing their immune roles such as phagocytosis, NF-κB–mediated transcription, production of chemokines and cytokines, degranulation, and other processes [37,38]. Moreover, fibrin deposits are routinely associated with bacterial foci within infected organ systems, with roles of both limiting the spreading of the pathogen and activating the immune response. Several experimental studies in animals with fibrinogen deficiency showed inefficient bacterial clearance following peritoneal infection with *Staphylococcus aureus* [39,40]. Several studies showed that serum fibrinogen is a valuable biomarker for predicting complicated acute appendicitis [41,42]. In a recent study by Bardakci et al. [43], a value of more than 564.5 mg/dL was considered an indicator of urgent need for operation in patients with acute cholecystitis.

Moreover, our findings showed that a fibrinogen value of ≥570 mg/dL, NLR ≥ 6.2, thickness of the gallbladder wall ≥ 6 mm, and transverse gallbladder diameter of ≥ 35 mm were associated with increased risk of DLC and may be a valuable tool in preoperative evaluation.

Imaging parameters were found to be important in predicting DLC. Several studies found a good correlation between DLC and several ultrasound parameters, such as gallbladder wall thickening, presence of pericholecystic fluid/abscess, gas in the wall or lumen, intraluminal membranes, irregular or absent gallbladder wall, and pericholecystic inflammation [25,44,45,46].

In a study by Serioka et al. [47], a defect of the cystic duct in magnetic resonance cholangiopancreatography was an independent predictor for bailout procedures [47]. Multiple CT parameters are highly associated with DLC, including irregular or absent walls, pericholecystic fluid, fat hyperdensity, thickening of the wall  >  4 mm, and the presence of hydrops [25]. Maehira et al. [48] described a novel parameter that proved to be relevant to discriminate difficult cases of LC based on the CT attenuation ratio of the arterial phase (ARAP), defined as the ratio of the maximum attenuation value of segment 5 to that of segment 8. The study found that among three-phasic dynamic CT findings, an ARAP ≥ 1.55 could be a predictive factor of difficult LC [48].

Both CT and MRCP examination of the hepatobiliary region offer valuable information in preoperative assessment of the anatomic variations of the biliary system, condition of cystic ducts, or perforation of the gallbladder wall. However, in many surgical departments, these exams are not routinely available for patients admitted in emergency for acute cholecystitis. Our protocol includes preoperative ultrasound evaluation in these cases. We found that a gallbladder wall thickness of 6 mm or more, transverse diameter of the gallbladder of more than 35 mm, the presence of double contour, and pericholecystic fluid were significantly associated with DLC. There is evidence regarding the predictive value of ultrasound exams for DLC in acute cases. However, thickness of the gallbladder wall of more than 4–5 mm was a significant predictor for difficult cases in other studies, including elective LC [45,49].

Predicting DLC is still a challenging issue in clinical practice, as independent parameters do not offer satisfactory sensitivity and specificity. Combining clinical, biological, and imaging data is important in preoperative evaluation. In the present research, we used the Tongyoo score to evaluate prediction of DLC in a study group with emergency LC for acute cholecystitis, with encouraging results.

Ramirez-Giraldo et al. [12] comparatively evaluated the accuracy of seven scoring systems in predicting DLC in the same group of patients with that underwent LC, either elective or in emergency. The study found three scores with comparable accuracy, with an AUC ROC varying from 0.761–0.783 [11,16,17]. The best value was achieved by the score of Tongyoo et al. [11], which included five clinical parameters (age > 50 years, male sex, history of previous biliary inflammation, obesity, grading by BMI, and history of previous abdominal surgery) and four ultrasound elements (contracted gallbladder, thickness of the gallbladder wall of 4 mm or more, pericholecystic collection, and impacted stones). Gallbladder wall thickness was an element present in all three scores, but the cut-off value varied between 3 mm [47] and 4 mm [46], and emergency presentation added an element of gravity. Comparison among different scoring systems may be somehow limited by different criteria used to define DLC in clinical studies. Further research is needed to obtain a wider consensus regarding the definition of difficult cholecystectomy.

The present study has some limitations. It is a retrospective study that was performed in a single center. Another element is that we could not differentiate between difficult LC related to sclerosis and severe local inflammation in our analysis.

## 5. Conclusions

Difficult cholecystectomies are associated with higher postoperative morbidity and increased hospital stay. The predictive factors for DLC should be acknowledged and assessed preoperatively for a better therapeutic approach. This study provides evidence of the role of fibrinogen as a valuable biomarker for DLC in acute cholecystitis, together with ultrasound evaluation of the gallbladder wall. Our research also found that the Tongyoo score is useful in preoperative risk assessment of emergency laparoscopic cholecystectomy for acute cholecystitis, with good specificity for DLC.

## Figures and Tables

**Table 1 diagnostics-14-00346-t001:** Clinical and biological data of the patients included in the study groups.

Variable	Non-Difficult LC Group (*N* = 150)	Difficult LC Group (*N* = 105)	*p*-Value
Age (mean ± SD, years)	52.4 (±16.2)	57.1 (±14.2)	0.009 *
Sex M (*n*, %) W (*n*, %)	41 (27.3%) 109 (72.6%)	43 (40.9%) 62 (59%)	0.03 *
BMI > 25 (*n*,%)	66 (44%)	57 (54.2%)	NS
Previous upper abdominal surgery (*n*,%)	9 (6%)	7 (6.6%)	NS
Diabetes mellitus (*n*,%)	17 (11.3%)	24 (22.8%)	0.02 *
ASA risk grading (*n*,%) I II III IV V	10 (6.6%) 86 (57.3%) 46 (30.6%) 7 (4.6%) 1 (0.6%)	3 (2.8%) 54 (51.4%) 42 (40%) 5 (4.7%) 1 (0.9%)	NS
Onset > 72 h (*n*, %)	63 (42%)	59 (56.1%)	0.03 *
Fever (*n*,%)	4 (2.6%)	13 (12.3%)	0.004 *
Jaundice (*n*, %)	17 (11.3%)	17 (16.1%)	NS
Murphy sign positive (*n*, %)	92 (61.3%)	82 (78.1%)	0.007 *
Leukocytes (cells/µL) (mean ± SD)	9622.5 (±4170.5)	12,731.6 (±7216.8)	<0.001 *
Leukocytes > 16,000/µL (mean ± SD)	7 (4.6%)	21 (20%)	<0.001 *
Neutrophils (cells/µL) (mean ± SD)	6639.5 (±1347.1)	9637.3 (±1527.7)	0.001 *
Lymphocytes (%) (mean ± SD)	1924.5 (±1.154.7)	2087.8 (±1489.5)	0.002 *
NLR (mean ± SD)	6.2 (±7.8)	11 (±25)	0.001 *
Thrombocytes (µL) (mean ± SD)	247,622.6 (±81,254.2)	264,731.4 (±174,673.6)	NS
Fibrinogen (mg/dL) (mean ± SD)	409.8 (±127.3)	574.5 (±238.7)	<0.001 *
INR (UI/L) (mean ± SD)	1.2 (±0.3)	1.3 (±1.2)	NS
AST (UI/L) (mean ± SD)	54.6 (±86.4)	57.5 (±99.9)	NS
ALT (UI/L) (mean ± SD)	74.8 (±90.6)	70.8 (±89.4)	NS
Total Bilirubin (mg/dL) (mean ± SD)	1.0 (±1.3)	1.2 (±1.8)	NS
Creatinine (mg/dL) (mean ± SD)	0.8 (±0.3)	0.9 (±0.4)	NS
TG13/TG18 Grading of AC I II III	81 (54%) 55 (36.6%) 14 (9.3%)	28 (26.6%) 48 (45.7%) 29 (27.6%)	<0.001 *
ERCP (pre/postoperative)	7 (4.6%)	9 (8.5%)	NS
Tongyoo score	2.9 ± 1.4	6.0 ± 2.6	<0.001 *

Footnote: * statistically significant (*p* < 0.05); NS—*p*-value > 0.2; ASA—The American Society of Anesthesiologists (ASA) physical status classification system; NLR: neutrophil-to-lymphocyte ratio; CBD: common bile duct.

**Table 2 diagnostics-14-00346-t002:** Imaging data of the patients included in the study.

Variable	Normal LC (*N* = 150)	Difficult LC (*N* = 105)	*p*-Value
CBD (mm) (mean ± SD)	5 (±2.2)	5 (±2)	NS
Dilatation of intrahepatic bile ducts (*n*, %)	12 (8%)	7 (6.6%)	NS
Transverse diameter of GB (mm) (mean ± SD)	28.7 (±7.9)	36.4 (±12.3)	<0.001 *
Length of GB (mm) (mean ± SD)	79.4 (±18.6)	93.3 (±27.1)	<0.001 *
Thickness of GB wall (mm) (mean ± SD)	4.2 (±1.5)	6 (±2.8)	<0.001 *
GB wall > 5 mm (*n*, %)	49 (32.6%)	74 (70.4%)	<0.001 *
The double contour of the GB wall (*n*, %)	27 (18%)	51 (48.5%)	<0.001 *
Types of GB lithiasis (*n*, %): GB stones Microlithiasis Sludge	109 (72.6%) 44 (29.3%) 18 (12%)	73 (69.5%) 32 (30.4%) 26 (24.7%)	NS NS 0.01 *
CBD lithiasis (*n*, %)	7 (4.6%)	3 (2.8%)	NS
Pericholecystic fluid (*n*, %)	4 (2.6%)	23 (21.9%)	<0.001 *
Ultrasound MURPHY sign + (*n*, %)	4 (2.6%)	14 (13.3%)	0.002 *

Footnote: * statistically significant (*p* < 0.05); NS—*p*-value > 0.2; +: positive.

**Table 3 diagnostics-14-00346-t003:** Surgical treatment and postoperative outcomes.

Variable	Normal LC *N* = 150	Difficult LC *N* = 105	*p*-Value
Surgery type (*n*, %): LC Conversion to OC	150 (100%) 0 (0%)	81 (77.1%) 24 (22.8%)	<0.001 *
Mean operation time (min, mean ± SD)	101.5 ± 43.6	213 ± 163.4	<0.001 *
Mean blood loss (mL, mean ± SD)	38.7 ± 43.1	273.4 ± 354.7	<0.001 *
Associated ERCP (pre/postoperative) (*n*, %)	7 (4.6%)	8 (7.6%)	NS
Total hospital stay (days, mean ± SD)	6.4 (±4.1)	8.1 (±4.1)	<0.001 *
Postoperative hospital stay (days, mean ± SD)	2.8 (±2.3)	4.3 (±2.7)	<0.001 *
Postoperative complications (*n*, %)	23 (15.3%)	35 (33.3%)	<0.001 *
• Bile leak	1 (0.6%)	3 (2.8%)	
• Main BDI	1 (0.6%)	2 (1.9%)	
• Hemorrhage	3 (2%)	6 (5.7%)	
• Intestinal injury	1 (0.6%)	0	
• SSI	1 (0.6%)	7 (5.7%)	
• Pleuro-pulmonary (pneumonia, pleural effusion)	3 (2%)	3 (2.8%)	
• Cardiovascular (malign hypertension, hypotension, hemodynamic instability)	5 (3.3%)	4 (3.8%)	
• Nosocomial infections	5 (3.3%)	6 (5.7%)	
• Others (upper gastrointestinal hemorrhage, anaphylactic shock, acute limb ischemia, acute pancreatitis)	3 (2%)	4 (3.8%)	
Death (*n*, %)	2 (1.3%)	1 (0.9%)	NS

Footnote: * statistically significant (*p* < 0.05); BDI = bile duct injury; SSI = surgical site infection.

**Table 4 diagnostics-14-00346-t004:** Postoperative complications according to Clavien–Dindo classification.

Postoperative Complications	Normal LC	Difficult LC	*p*-Value
Grade 1 (SSI)	1 (0.6%)	7 (6.6%)	0.006 *
Grade 2 (surgery-related complications, treated pharmacologically)	4 (2.6%)	8 (7.6%)	0.03 *
Grade 3 (surgery-related complications requiring surgical, endoscopic, and radiological treatment)	1 (0.6%)	6 (5.7%)	0.01 *
Grade 4 (general complications, requiring intensive care)	8 (5.3%)	9 (8.5%)	NS
Grade 5 (death)	2 (1.3%)	1 (0.9%)	NS

Footnote: * statistically significant; NS—*p*-value > 0.2.

**Table 5 diagnostics-14-00346-t005:** Multivariate analysis results for DLC—model 1.

	Odds Ratio	*p*-Value
**Intercept**		
	0.00624 [0.00153; 0.0254]	<0.0001 ****
**Fibrinogen**		
Risk for each 1-unit increase	1.004 [1.002; 1.005]	0.00207 **
**Transverse diameter of gallbladder**		
Risk for each 1-unit increase	1.05 [1.02; 1.08]	0.00393 **
**Thickness of gallbladder wall**		
Risk for each 1-unit increase	1.4 [1.18; 1.67]	1.38 ×10^−4^ ***

** *p* < 0.01, *** *p* < 0.001, **** *p* < 0.0001.

**Table 6 diagnostics-14-00346-t006:** Predictive value of the tested variables.

Parameter	AUC	Cut-Off Value	Sensitivity	Specificity	Positive Predictive Value	Negative Predictive Value
Tongyoo score	0.857	>4	67.6%	88%	79.7%	79.3%
Regression model	0.802	>0.5	60.5%	86.6%	75.7%	75.6%
Fibrinogen (serum)	0.701	>536	53.3%	86.7%	73.5%	72.7%
NLR	0.634	>6.33	49.5%	77.3%	60.2%	68.7%
GB wall thickness	0.734	>4.7	70.5%	67.3%	59.9%	76.6%
TG 13/18	0.659	>1	72.3%	53.3%	51.8%	73.4%
L > 16,000	0.577	>0	20.0%	95.3%	73.5%	63.1%
Total Leukocytes	0.663	>13,200	38.1%	85.3%	63.3%	66.4%
Pericholecystic fluid	0.596	>0	21.9%	97.33%	84.9%	64.1%

## Data Availability

The data presented in this study are available upon request from the corresponding author. The data are not publicly available due to privacy concerns.

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
