# Peer review of "Predictive Factors for Difficult Laparoscopic Cholecystectomies in Acute Cholecystitis"

_diagnostics, 2024, doi:10.3390/diagnostics14030346_

Round 1

Reviewer 1 Report

Comments and Suggestions for Authors

The topic of the paper is interesting and covers a theme which is frequently faced within the HBP surgical issue. 

but  There are significant limitations to using inflammatory markers and ultrasound findings to predict difficult LC procedures. 

You need to  revise and add relevant other factors of the prediction of surgical difficulty  and to summarize the table contents more.

There are a lot of researchs on predictive models for laparoscopic cholecystectomy in acute cholecystectomy. Some inflammatory factors are one of the important factors, but imaging such as CT or MRCP is really important.
(need to evluate variation of biliary system, condition of cystic duct, perforation of GB.....) 

This study presented only ultrasound images, but they are quite limited to evaluate. When evaluating DLC, intraoperative factors such as operartiver times, estimated blood loss, and port number should be included, but the author do not present.

The author needs to revise and add above.

Comments on the Quality of English Language

Minor editing of English language required.

Author Response

Dear reviewer,

Thank you for your time spend reviewing our work and your valuable comments. We have now revised the manuscript according to your recommendations:

We added the required information about the comparative operative times and estimated blood loss. Also, in the Materials and methods section, we described more details about the perioperative preparation, surgical technique, including the port number, as required.

We have revised the statistics and compared our regression model with another predictive scoring system of Tongyoo et al with good predictive value, according to the following references:

Tongyoo A, Chotiyasilp P, Sriussadaporn E, et al.. The pre-operative predictive model for difficult elective laparoscopic cholecystectomy: a modification. Asian J Surg 2021;44:656–661.We have revised the tables and condensed the data presented as required.

This will be the first testing of this scoring system on a group of patients with acute cholecystitis. Previous studies evaluated both elective and emergency LC.

We totally agree with your point of view that CT exam and MRCP offers more valuable information regarding the detailed anatomy of the hepatobiliary region and are extremely useful in preoperative evaluation in challenging situation. We added a paragraph with this idea in the discussion section.

 However, these investigations are not available by routine for patients admitted in emergency with acute cholecystitis. The aim of the present paper is to offer a simple tool, based on some routine investigations that could raise a red-flag for possible intraoperative difficulties in laparoscopic cholecystectomies for acute cholecystitis.

We have carefully revised English language, as required.

We hope in this revised version, you will find our paper suitable to be published.

Kind regards,

Prof. Dr. Dragos Serban

Reviewer 2 Report

Comments and Suggestions for Authors

Comments to Authors:
This submission by Stoica and colleagues from Bucharest addresses the topic of predicting the potential trial severity of a laparoscopic cholecystectomy for patients with acute cholecystitis.

Major Comments:

1. In the  it would be nice to give age of the positive predictive values as well as the sensitivity. I have not read the remaining part of your manuscript yet BUT these values I assume you will analyze and knowing the positive and negative predictive values of fibrinogen etc not all but several of the more dramatic values would stimulate the reader to continue reading

2. I would strongly urge you to shorten the 2nd paragraph of the introduction and the upfront k the third paragraph introduce the concept of preoperative identification of a patient with a potentially DLC that would be expected to help with patient safety-as written you focus on the DLC and not the PREDICTION of a DLC which is the focus of the paper

3. Line 101 how is lithiasis acute cholecystitis an exclusion factor? Did you mean non-lithiasis ? And if so why even exclude them? Honestly I would not use the term “lithiasic cholecystitis” while I know. What you mean this is not the common English phrase; better and more understood terms used more frequently are callous cholecystitis and calculus cholecystitis see also line 96.

4. Line 122 are you planning to calculate positive predictive values, negative predictive value, specificity, sensitivity, etc? If not why? These may prove especially important.

5. Lines 166-171. How are these data relevant?

6. Line 181 -182. I question this is it coming ally relevant 1 day difference?

7. I am disappointed Ted after reading the results section that you have not calculated positive, negative predictive values not specificity and sensitivity values for j stance if one calculates whether an increased WBC is predictive of acute appendicitis yes it will be but the specificity is far too low to use this as a solid diagnostic criterion

8. I am going to give you several suggestions to see are far too many statistical analyses presented in the tables why not stick to the really important and verifiable and clinically important statistical comparisons in the results section and then all the other really non important data as supplementary in supplementary data section. e.g. line 244-246 with NLR what are the values for an increased NLR sensitivity specificity positive and negative predictive ales?

9. Did you try to report e the reference and the parameter used by Tongyoo if. It why not are yours better easier to obtain more accurate? This would be an important contribution to the literature to do this- you might find that that study gives better results than yours

Minor comments

1. Line 32 define DCL
2. Line 32 too many decimal points with less than 1000 patients you only get 1

3. Line 57 change inappropriate to inadequate or hard to usual use

4 line 126 and Table 1 gender is the phenotype while sex is the genotype. I would suggest that you use genotype her because first this is a scientific paper and second that may very well be important ( as your abstract suggests)

5. Table one round off all the percentages, also I see you mention IQR with only one value. You need to talk to a statistician the IQR is not the SD and the IQR has two numbers sorry this raises my concern of your statistical analysis. Moreover you do not mention any median values or IQR in your stats section. This needs to be reviewed with one of your statisticians. You have a great idea but I am not convinced you understand these terms and the use of non parametric statistical analyses for instance table 1 AST IN COLUMN 1 you say ( mean, range, median). This is important because the median is a non appropriate way to present these data look at the difference in mean and median also how large the SD is; these and other data ( e.g. ALT, NLR, etc) are similar

6. Table 2 has similar problems

7. Table 3 you cannot statistically compare bile leak with only 1 and 3 patients again talk with a statistician

8. Lines 272 -284. What are the “higher odds “

9. Line 289-290 why not also say that a preoperative CT is not indicated? You should not say that you did not use CT for limited resources- No one uses CT routinely for acute cholecystitis it would not be appropriate.

Authors I had been quite critical with many of my suggestions, but they are offered I hope you understand in a constructive way. You have GREAT DATA . But there are scientific problems with your analysis AND the bottom line of what are the important data points and their combination that accurately predicts a DLC. That is what the Reader will want and doesn't get even in the conclusions section.

Comments on the Quality of English Language

very good usage of English I did make a few suggestionscc certain phrases and terms

Author Response

Dear reviewer,

Thank you for your time spend reviewing our work and your valuable comments. We have now revised the manuscript according to your recommendations:

We have revised the statistics and compared our regression model with the predictive scoring system of Tongyoo et al, as suggested, according to the following references:

Tongyoo A, Chotiyasilp P, Sriussadaporn E, et al.. The pre-operative predictive model for difficult elective laparoscopic cholecystectomy: a modification. Asian J Surg 2021;44:656–661.

This will be the first testing of this scoring system on a group of patients with acute cholecystitis. Previous studies evaluated both elective and emergency LC.

We have revised the tables and condensed the data presented as required.

  1. In the it would be nice to give age of the positive predictive values as well as the sensitivity. I have not read the remaining part of your manuscript yet BUT these values I assume you will analyze and knowing the positive and negative predictive values of fibrinogen etc not all but several of the more dramatic values would stimulate the reader to continue reading

R: We have improved our statistics according to your recommendations. We have added positive pred value, negative pred. value, sensibility and specificity for fibrinogen, gallbladder wall thickness, NLR, regression model and the scoring system of Tongyoo, as suggested.

  1. I would strongly urge you to shorten the 2nd paragraph of the introduction and the upfront k the third paragraph introduce the concept of preoperative identification of a patient with a potentially DLC that would be expected to help with patient safety-as written you focus on the DLC and not the PREDICTION of a DLC which is the focus of the paper

R: We have revised the introduction according to your recommendations.

  1. Line 101 how is lithiasis acute cholecystitis an exclusion factor? Did you mean non-lithiasis ? And if so why even exclude them? Honestly I would not use the term “lithiasic cholecystitis” while I know. What you mean this is not the common English phrase; better and more understood terms used more frequently are callous cholecystitis and calculus cholecystitis see also line 96.

R: Many thanks for your keen evaluation of our work. We have corrected, according to the recommendation. Indeed, we meant acalculous acute cholecystitis as an exclusion criteria, based on the fact that these patients are usually critically ill with atherosclerotic heart disease, recent trauma, burn injury, surgery, or hemodynamic instability. We considered that all these associated conditions may deeply impact the inflammatory biomarkers that we studied.

  1. Line 122 are you planning to calculate positive predictive values, negative predictive value, specificity, sensitivity, etc? If not why? These may prove especially important.

R: Thank you for the suggestion, we have included these data in the revised version.

  1. Lines 166-171. How are these data relevant?

We agree this paragraph is not relevant for the aim of our study and decided to remove it, for the fluency of the paper.

  1. Line 181 -182. I question this is it coming ally relevant 1 day difference?

R: We agree that while statistically significant, the differences in total hospital stay and postoperative stay have a limited clinical significance. We added this comment.

  1. I am disappointed Ted after reading the results section that you have not calculated positive, negative predictive values not specificity and sensitivity values for j stance if one calculates whether an increased WBC is predictive of acute appendicitis yes it will be but the specificity is far too low to use this as a solid diagnostic criterion

R: Thank you for the comment. We totally agree your point of view. We have revised the statistics according to your recommendations.

  1. I am going to give you several suggestions to see are far too many statistical analyses presented in the tables why not stick to the really important and verifiable and clinically important statistical comparisons in the results section and then all the other really non important data as supplementary in supplementary data section. e.g. line 244-246 with NLR what are the values for an increased NLR sensitivity specificity positive and negative predictive ales?

R: According to your comments, we have shortened our statistics and sticked to the most relevant data for the reader. We added the sensitivity, specificity, positive prediction and negative prediction for the most relevant parameters, as suggested.

  1. Did you try to report e the reference and the parameter used by Tongyoo if. It why not are yours better easier to obtain more accurate? This would be an important contribution to the literature to do this- you might find that that study gives better results than yours

R: Thank you for the suggestion. We have revised the manuscript and calculated the score of Tongyoo, based on our statistical data and compared the results.

Minor comments

  1. Line 32 define DCL

R: We have corrected.

  1. Line 32 too many decimal points with less than 1000 patients you only get 1

R: I am not sure if I correctly understood, do you mean the p value? We have corrected and kept the first significant decimal.

  1. Line 57 change inappropriate to inadequate or hard to usual use

R: We have corrected.

4 line 126 and Table 1 gender is the phenotype while sex is the genotype. I would suggest that you use genotype her because first this is a scientific paper and second that may very well be important ( as your abstract suggests)

Thank you, we have corrected.

  1. Table one round off all the percentages, also I see you mention IQR with only one value. You need to talk to a statistician the IQR is not the SD and the IQR has two numbers sorry this raises my concern of your statistical analysis. Moreover you do not mention any median values or IQR in your stats section. This needs to be reviewed with one of your statisticians. You have a great idea but I am not convinced you understand these terms and the use of non parametric statistical analyses for instance table 1 AST IN COLUMN 1 you say ( mean, range, median). This is important because the median is a non appropriate way to present these data look at the difference in mean and median also how large the SD is; these and other data ( e.g. ALT, NLR, etc) are similar

R: Thank you for your valuable comments. After the revision with our statistician, we decided to keep in the table the comparative mean+/-DS, as you recommended, so that the tables could be easier to read and follow by the reader. Independent t-test or Mann–Whitney test was applied for all quantitative variables, the according to data distribution (e.g. normally distributed or not), while Chi-square test or Fisher's exact test was used for all categorical variables.

  1. Table 2 has similar problems

R: we have revised it in a similar manner.

  1. Table 3 you cannot statistically compare bile leak with only 1 and 3 patients again talk with a statistician

We totally agree that some of the postoperative complications are too few to be compared for our study groups. For this reason, we kept the p value only for the total number of complications, and removed the comparative evaluation for each complication taken separately.

  1. Lines 272 -284. What are the “higher odds “

We have rephrased it, thank you for the correction.

  1. Line 289-290 why not also say that a preoperative CT is not indicated? You should not say that you did not use CT for limited resources- No one uses CT routinely for acute cholecystitis it would not be appropriate.

We have corrected, according to the suggestion.

Thank you again, for all your useful comments that helped us a lot to improved our work! We do hope in this revised version you will find our paper suitable for publication.

Kind regards,

Prof. dr. Dragos Serban

Round 2

Reviewer 1 Report

Comments and Suggestions for Authors

Thanks for your effort.

I don't have any comments anymore in  your revised manuscript.

I hope this study will be contribute to this journal.

Author Response

Dear reviewer,

Thank you very much for your helpful comments and kind appreciation of our work.

Best regards,

Prof. Dr. Dragos Serban

Reviewer 2 Report

Comments and Suggestions for Authors Comments to the authors: This is a re-review of a submission by drs Stoica et al from Carol Davila, Ovidious, and University hospitals on the topic of risk factors for lap cholesterol on patients with Acute cholecystitis. I have read the entire submission and several of my comments are very easy to fix (mostly grammar).

Major comments: 1. Table 1 INR. That CANNOT be statistically significant and certainly is not clinically significant drop the p value. My strong suggestion is to put NS (not significant) rather than the calculated value for all p values >0.2 here and in table 2 and 3. The Tongyoo score should be 6.0 ± 2.6. Not just 6. Also it is late to fix this but I. Truth when the SD is so. Ery large llome for WBC, lymphocytes, platelets, these values would have been beat to be presented as median values with interquartile ranges and compared with a non parametric statistical analysis - but for this paper I would leave it as is 2. lines 243-244 it might be nice to give the ranges of positive predictive values for these individual blood tests because I would bet that the positive predictive. Amie’s for these would be less than 80% thus their individual importance (positive and negative as well as sensitivities and specific values are too low to really use as an absolute value.
3. Lines 297-301. Well done!!!!!! That is the real world!

Minor comments: Authors please note I can only imagine how difficult it is to write in a non primary language so all the following suggestions are not crucial but will help the readability of your excellent submission - so please take these as constructive ok not destructive criticism! 1.line 28 hangs “adherences” to “suspected adherence to and involvement of adjacent important structures” 2. Line 57 add the word “a” before “”difficult” 3. Line 64 replace “on” with “involving” 4. Line 70 , 73 , 123, 144, 174, 283, and 310 change “imagistic” to “imaging” 5 line 80 define WSES 6. Line 85 add the phrase “patients specifically with” before the word “acute” 7. Line 100 change to “ empire of the gallbladder” instead of proctologists 8. Line 105. Replace hidroelectolitis rebalance” to “correction of abnormalities in electrolyte concentrations” 9. Line 112 I would add “Postoperative drainage of the gallbladder fossa” instead of just “drainage” 10. Line 119 change angiocholitis to cholangitis 11. Line 147. Sensibility needs to be changed to “sensitivity” here and throughout the text 12. Line 169 please limit the thickness of the gallbladder wall to 6.1 and 4.2. You cannot measure wall thickness with an accuracy of one hundredth of a mm 13. Table 3 and text round off the % for the postop complications line. 15.33 to 15.3 and 33.33 to 33.3% Also when comparing the individual complications because the absolute numbers are small do not show percentages they are meaningless with such small numbers also this is much more scientifically and statistically correct! 14. Line 184 delete adherences and change to “inability to safely dissect the important structures ( bile duct, cystic duct, hepatic arteries, or portal venous structures) from surrounding tissues 15. Line 184. For postop stay limit to 1.7 and 1.5. Hundredths of a day is meaning less 16. Line 192 delete “ we’re significantly” to “appeared to be.; the comparison of 1 in 150 to 7 in 105 is a. It soft 17. Line 233 I would suggest that you consider replacing the end of the sentence starting with ”LC performed in an emergency” to “ patients undergoing LC for acute cholecystitis” 18. I see later in the text that you are still defining DLC. No need to once you have defined this abbreviation you do not need to define it again
19. Line 280 add the phrase “our findings of “ after the word “Moreover” here you can stress YOUR STUDY!

Comments on the Quality of English Language

The English is quite good. I made a number of suggestions cc word choice. You did a great job, yes it needs some help but overall well done!

Author Response

Dear Reviewer,

Many thanks for your kind support and valuable comments, that helped us to improve our manuscript.

We have carefully revised the manuscript according to your suggestions:

Major comments: 1. Table 1 INR. That CANNOT be statistically significant and certainly is not clinically significant drop the p value. My strong suggestion is to put NS (not significant) rather than the calculated value for all p values >0.2 here and in table 2 and 3. The Tongyoo score should be 6.0 ± 2.6. Not just 6.

R: Thank you for observing, indeed it was 0.2, not significant. We have corrected. We also had corrected p value in all table and put NS when the calculated value was p>0.2 as recommended.

We have corrected in the value for Tongyoo score, as recommended.

 Also it is late to fix this but I. Truth when the SD is so. Ery large llome for WBC, lymphocytes, platelets, these values would have been beat to be presented as median values with interquartile ranges and compared with a non parametric statistical analysis - but for this paper I would leave it as is.

R: Thank you for the observation. We totally agree with your comment, indeed there is quite a large variation. Actually, we have also calculated median values and interquartile rages (IQR) for all these parameters in the beginning. However, as we received the recommendation to condense the data presented in tables, we chose mean+/-SD, as we suppose the readers would be more familiar with these. We added a Supplementary file S1 with all calculated data, including median and IQR, for those who might be interested in more statistics.

We also used Mann-Whitney U test, a non-parametric test, for statistical analysis in these cases. We added this info in the Materials and Methods.

  1. lines 243-244 it might be nice to give the ranges of positive predictive values for these individual blood tests because I would bet that the positive predictive. Amie’s for these would be less than 80% thus their individual importance (positive and negative as well as sensitivities and specific values are too low to really use as an absolute value.

R: We added the PPV and NPV for other blood tests and imagistic parameters in table 7, as suggested. Indeed, your presumption was right, their individual importance as predictive value is too low for clinical practice. We added this finding.

  1. Lines 297-301. Well done!!!!!! That is the real world!

R: Thank you!

Minor comments: Authors please note I can only imagine how difficult it is to write in a non primary language so all the following suggestions are not crucial but will help the readability of your excellent submission - so please take these as constructive ok not destructive criticism!

R: Thank you so much for all your precious comments that improved the readability of our paper. We have done all the required corrections, exactly as indicated.